# Towards Detecting Social Events by Mining Geographical Patterns with VGI Data

**Zhewei Liu**, **Xiaolin Zhou, Wenzhong Shi * and Anshu Zhang**

Department of Land Surveying and Geo-informatics, The Hong Kong Polytechnic University, Hung Hom 999077, Hong Kong, China; jackie.zw.liu@connect.polyu.hk (Z.L.); xiaolin.zhou@connect.polyu.hk (X.Z.); anshu.zhang@connect.polyu.hk (A.Z.)

* Correspondence: john.wz.shi@polyu.edu.hk; Tel.: +852-9348-6920

**Abstract:** Detecting events using social media data is important for timely emergency response and urban monitoring. Current studies primarily use semantic-based methods, in which "bursts" of certain semantic signals are detected to identify emerging events. Nevertheless, our consideration is that a social event will not only affect semantic signals but also cause irregular human mobility patterns. By introducing depictive features, such irregular patterns can be used for event detection. Consequently, in this paper, we develop a novel, comprehensive workflow for event detection by mining the geographical patterns of VGI. This workflow first uses data geographical topic modeling to detect the hashtag communities with VGI semantic data. Both global and local indicators are then constructed by introducing spatial autocorrelation measurements. We then adopt an outlier test and generate indicator maps to spatiotemporally identify the potential social events. This workflow was implemented using a real-world dataset (104,000 geo-tagged photos) and the evaluation was conducted both qualitatively and quantitatively. A set of experiments showed that the discovered semantic communities were internally consistent and externally differentiable, and the plausibility of the detected events was demonstrated by referring to the available ground truth. This study examined the feasibility of detecting events by investigating the geographical patterns of social media data and can be applied to urban knowledge retrieval.

**Keywords:** event detection; volunteered geographic information; geographical pattern mining; feature transformation

## 1. Introduction

Volunteered geographic information (VGI) [1] is a recent development that has considerably influenced the way humans interact and information is retrieved. Semantic as well as location information can be attached to VGI, providing new opportunities for research on human mobility and urban systems. Recent observations have also found that some emerging news and events spread rapidly throughout social media, making social media a new tool for event detection and disaster response [2].

Currently, most research on event detection with social media uses semantic-based methods and text mining [3,4]. Platforms, such as Twitter, provide accessible streaming data, which contains real-time text information, making it possible to detect emerging social events by examining the change in semantic-based features [5–7]. However, a new event detection methodology is proposed by investigating the geographical patterns of VGI data. Our intuition is that a social event will affect how the objects spatially distribute across certain regions and how they mutually interact thus causing irregular geographical patterns, especially irregular human mobility and interaction patterns (e.g., sports games causing intense human aggregation or terrorism attacks causing sudden evacuation from certain regions). By introducing depictive measuring features with VGI data and identifying the

feature irregularity, such geographical patterns, in turn, can be used to distinguish potential social events. To the best of our knowledge, previous works mainly detected social events by word frequency and semantic analysis, and there is still a need for a comprehensive methodology to effectively detect events by investigating irregular geographic patterns.

Consequently, a major motivation for this research is to provide new insights into the following issues: what kinds of features can be used for detecting events, or from what aspects can a social event be differentiated with VGI data. Specifically, the novelty and point of our research was to detect social events by mining the geographical patterns of VGI and using geography-based features. To do so, we proposed a comprehensive workflow that consists of three parts: (1) semantic community discovery, (2) geography-based event representation, and (3) irregular event feature detection. First, a data-driven geographic topic modeling method was used to detect hashtag communities and identify social media topics. Second, by introducing quantitative spatial autocorrelation indicators, a novel global index was constructed for representing the potential events in the created feature space. A time series of the univariate event feature was then generated with variable temporal granularities. Third, an outlier test was adopted to detect event feature irregularity, and the global index was localized for finding event location. A detailed event description can then be obtained using the detected semantic and spatiotemporal identification. We conducted the experiment with a real-world dataset (104,000 geo-tagged photos) and verified the effectiveness of the proposed workflow both qualitatively and quantitatively.

The main contributions of this paper are as follows:

- We presented a novel workflow to take advantage of new geographical features for event representation and used the change ("outliers") in the geographical patterns revealed by VGI to differentiate potential events.
- We developed an indicator system that includes both global and local indicators for pinpointing social events, as well as a comprehensive data model for event detection, with the consideration of semantic and geographical identification.

The remainder of the paper is organized as follows: Section 2 summarizes the related work, Section 3 presents the proposed workflow, Section 4 reports the experimental results, Section 5 discusses the performance of our topic modeling and event detection methods, and finally, Section 6 concludes the paper.

## 2. Related Work

Our workflow detects social events by investigating the semantic topics and geographic patterns of VGI data. In this part, previous related works on geographical topic discovery and event detection with social media are presented.

Geographical topic discovery [8] finds topics in spatial big data (e.g., GPS-associated documents, geo-tagged social media data) in a geographical context. Among the various geographical topic modeling methods, statistical models such as PLSA [9] and Latent Dirichlet Allocation [10] are commonly used for geographic topic modeling [11–14]. However, these methods have certain limitations in relation to social media data because short noisy texts are predominant and a priori knowledge, such as a predefined count of topics, is unavailable [15]. In other research studies, other forms of information have been used, such as images [16] and heterogeneous unstructured articles [17], the correlation of user movement and interest [18], and geographical diversity and interest distribution of users [19], to facilitate the process of topic discovery. In our work, to detect thematic meanings, we used a data-driven geographic topic modeling method based on the assumption that the semantic information attached to a social media post, such as hashtags, is quite self-explanatory and thus, the potential topics can be detected. The method is data-driven and does not require training data or a priori knowledge such as topic counts, thus reducing potential perceptual biases.

Another research area related to our work is event (anomaly) detection with social media data. A challenging problem is to extract useful, structured representations of events from the disorganized corpus of noisy posts [20]. Sudden increases in the frequency ("bursts") of sets of keywords have been popularly used for detecting new events in a data stream [21]. Some other approaches include wavelet-based clustering of frequency signals, topic clustering with meta-data analysis and domain-specific approaches [22,23]. In Schinas et al. [24], the authors grouped the event detection methods into three categories: (1) feature-pivot, i.e., detection of abnormal patterns in the appearance of features [5,23,25,26]; (2) document-pivot, i.e., clustering documents with similarity measures [27–30]; and (3) topic modeling, i.e., using statistical models to identify events [31–34]. Becker et al. [35] highlighted four feature categories (temporal, social, topical, and Twitter-centric features) that can be used for modeling event features. These methods are mainly semantic-based, that is, they use text mining techniques to investigate the change in social media semantic information for event detection. Other studies have used spatial-related methods. These have addressed issues such as measuring regional irregularities with post count and user count [36], using content-based methods to detect event location [37,38], estimating the influenced area of an event with kernel density estimation [39] investigating the geographic extent to find localized events [40], assessing the impact area of a natural disaster [41], and detecting events (outliers) by finding the weighted centroids outside the spatial standard deviation ellipse [42]. Compared with semantic-based methods, these spatial-related methods basically focused on identifying the spatial information of the detected event or using the change in the frequency of the posts/users in certain spatial regions to detect events. In our work, we postulate that social events can not only lead to "bursts" in certain semantic signals and the number of posts/users but they also cause structural change in geographical patterns, such as how objects spatially distribute and interact with each other. So, rather than investigating the numeric change in posts/users, we tried to construct features to indicate the structural change in objects' geographical distribution patterns. Specifically, we picked up the clustering patterns (e.g., spatial autocorrelation) and designed the event features accordingly. To the best of our knowledge, detecting social events by mining irregularities in geographical patterns has not been adequately investigated by previous studies, and this work can shed some light on relevant fields.

## 3. Proposed Methods

In this section, the data model and analytic workflow of our methods are presented, followed by a detailed description of each data handling procedure. First, we introduce a geographic topic modeling method to investigate the semantic meaning of social media posts. Next, we explain a novel event feature representation method based on the geographical patterns of the social media post distribution. Finally, we describe the method for irregularity detection and location indication based on the data model.

### 3.1. Data Model and Analytic Workflow

In our model, to detect social events, the following issues were considered:

- What are the social events about? In other words, semantically, how can we identify the potential topics of the social events using social media data?
- How should the social event be depicted from a geographical perspective? Where is the event location? Is there any irregular geographical pattern (e.g., crowd aggregation, evacuation) caused by the event, and how can such patterns be represented with the geo-tagged social media posts?

Based on the above considerations, our event detection model was defined as follows:

$$E = W\,(S,\ G) \tag{1}$$

where $E$ is the event detection result, $S$ is the semantic identification to indicate the event topics, $G$ is the geographical patterns of human mobility revealed by VGI, and $W$ is the analytic workflow.

The underlying perception is that a social event may affect regular human mobility and interaction patterns. By introducing features derived from quantitative measurements of such patterns, a new feature space can be created, and the social event can be represented and detected with the newly created features.

As represented in Equation (1), the semantic and geographical patterns are both considered in the model. Thus, to address the semantic and geographical issues, we constructed a three-fold analytic workflow (Figure 1), which consists of three interrelated modules: (1) semantic community discovery; (2) geography-based event representation; and (3) detection of event feature irregularity.

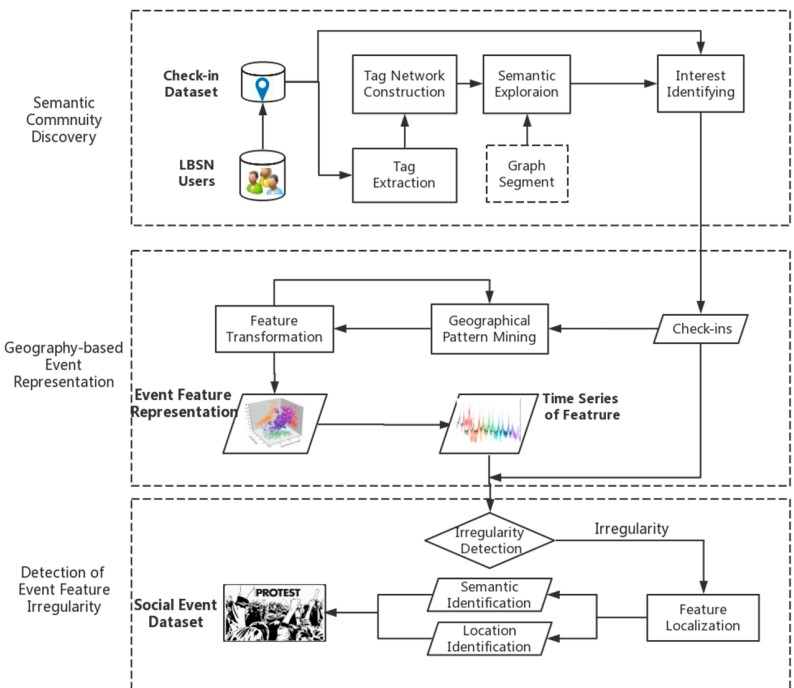

**Figure 1.** Proposed analytic workflow.

Semantic Community Discovery: In this step, the semantic content attached to the VGI is investigated first to determine the user activity associated with the geo-tagged posts (e.g., geo-tagged photos). This step starts by retrieving geo-tagged media data from a relevant API (check-in dataset). Furthermore, the hashtags in the titles attached to the geo-tagged photos are extracted, and based on this an undirected "hashtag network: model is built, where each hashtag is denoted as a network node, and the co-occurrence frequency between hashtags is assigned to the weight of the edge connecting the corresponding nodes in the 'hashtag network' (Tag Network Construction in Figure 1). A greedy optimization method is then implemented to discover the communities from the hashtag network, and a common topic is assigned to the hashtags within the same community (Semantic Exploration in Figure 1). The topics of geo-tagged photos are further identified by introducing the topics of attached hashtags as an indicator, as shown in Figure 2 (Interest Identifying in Figure 1).

Geography-Based Event Representation: In this step, a new event feature representation method is developed. By introducing quantitative spatial autocorrelation indicators (Geographical Pattern Mining), we constructed a global index for representing the potential event in the created feature space (Feature Transformation and Event Feature Representation). A time series of the univariate event feature with different temporal granularity can then be generated by adjusting the temporal range parameter.

Detection of Event Feature Irregularity: After mapping the geographical patterns into a new feature space, we adopt an outlier test to detect the feature irregularities. The global index is then localized for identifying the event location. By combining the semantic and location identification, a detailed description of the detected event can be obtained.

The details of each data model and module are presented below.

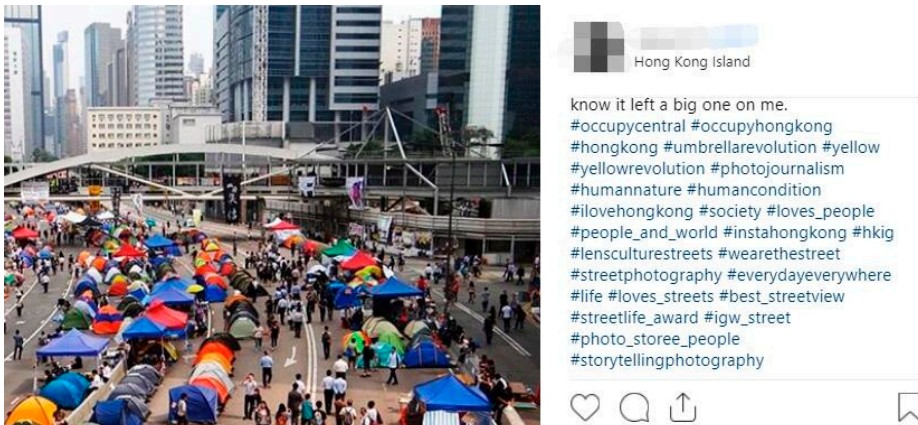

**Figure 2.** Photo sample from Instagram. The photo has a title with multiple hashtags (e.g., #occupycentral, #umbrellarevulotion, #best_streetview) to annotate, indicating potential topics for the photo.

### 3.2. Semantic Community Discovery

The semantic content attached to the geotagged social media posts is first investigated.

Statistical models such as Latent Dirichlet Allocation (LDA) [10] show certain limitations in handling social media data given that short noisy texts are predominant and a priori knowledge is unavailable [15]. Consequently, we developed an unsupervised data-driven method, which uses the co-occurrence pattern of the self-explanatory information for detecting semantic communities to investigate the semantic divisions of social media topics. The assumption for this method is that the semantic information, such as hashtags, is highly relevant to the topics of a social media post, and hashtags with similar semantic meaning tend to have a high probability of co-occurrence (as shown in Figure 2). Based on the above assumptions, a hashtag network model is built to investigate the geographic pattern of the social media topic in the rich-hashtag environment. The details of the method are defined below:

Let $P$ be a collection $P = [p_1, \dots, p_n]$ of social media posts $p_i = (l_i, t_i, tgs_i, tps_i)$, where $l_i$ is the location of post $p_i$, $t_i$ is the time-stamp, $tgs_i$ is the union of the hashtags attached to $p_i$, and $tps_i$ are the potential topics remaining to be identified. Let $HT = \bigcup_{i=1}^{n} tgs_i = [tg_1, \dots, tg_m]$ be the hashtag union of $tgs_i$, i.e., if and only if *item* $\epsilon$ $tgs_i$ ($i = 1, \dots, n$), *item* $\epsilon$ $HT$.

The co-occurrence function $CRC(p_i, tg_j, tg_k)$ is defined as follows:

$$CRC(p_i, tg_j, tg_k) = \begin{cases} 1, \ if \ tg_j \ \epsilon \ p_i.tgs_i \ \text{AND} \ tg_k \ \epsilon \ p_i.tgs_i \\ 0, \ otherwise \end{cases} \tag{2}$$

where $p_i$ is a post item from collection $P$, $tg_j$ and $tg_k$ are hashtag items from $HT$.

A hashtag network is further defined as $HTN = (V, E)$, where $V$ is a set of vertices, each of which denotes a hashtag $tg_i$ in $HT$, and $E$ is a set of undirected edges connecting the vertices. The weight of the edge is calculated as:

$$A_{jk} = \sum_{i=1}^{n} CRC(p_i, tg_j, tg_k) \tag{3}$$

where $A_{jk}$ is the weight assigned to the edge connecting vertices $j(tg_j)$ and $k(tg_k)$. The weight, calculated by summing the co-occurrence frequency of the corresponding pair of hashtags, represents the connectivity between nodes.

Based on the assumption that hashtags with similar semantic meaning have a high probability of co-occurrence and vice versa, we grouped the hashtag network into several communities based on their connectivity so that the vertices within the same community have a dense connection and share the same topic. In other words, a community in the network is interpreted as a group of hashtags sharing one common topic, and based on the connectivity between nodes, different topics can be detected.

The concept of modularity [43] is imported as a measurement of the strength of network division and connectivity. Modularity $Q$ is often used in optimization methods for detecting communities in a network and defined as follows [44]:

$$Q = \frac{1}{2m} \sum_{i,j} \left[ A_{ij} - \frac{k_i k_j}{2m} \right] \delta(c_i, c_j) \tag{4}$$

where $A_{ij}$ is the weight of the edge connecting vertices $i$ and $j$, $k_i = \sum_j A_{ij}$ is the sum of weight of edges linking to the vertex $i$, $m = \frac{1}{2} \sum_{i,j} A_{ij}$, $c_i$ is the community that vertex $i$ belongs, the δ-function $\delta(u, v)$ is 1 if $u = v$, and 0 otherwise.

To detect communities, we employed the Louvain algorithm [44]. As a greedy method inspired by the optimization of modularity, the Louvain algorithm is data-driven and does not require an a priori selection of the community count, which can detect network communities with fewer potential perceptual biases. The algorithm is applied to the hashtag network $HTN$ and several hashtag communities can be detected. Based on the semantic meaning of hashtags, a topic can be assigned to each community (as shown in Figure 3). The topics of social media post $p_i. tps_i$ can be identified by referring to the topics of $p_i.tgs_i$.

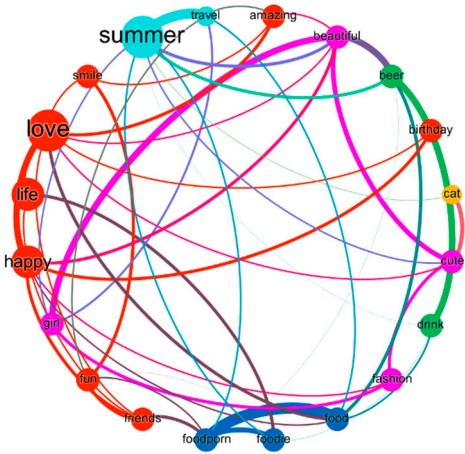

**Figure 3.** Semantic community discovery based on hashtag co-occurrence pattern and modularity, with node and edge size proportional to occurrence frequency, and the node color indicating community category.

### 3.3. Geography-Based Event Representation

After identifying the semantic topics of the geo-tagged posts, we constructed geography-based features for event representation. We consider that a social event may lead to irregular human mobility (e.g., crowd aggregation), and such irregularity can be revealed by the geographical pattern of VGI. Consequently, a geography-based event representation method was proposed. The spatial autocorrelation is introduced as a geographical description of human mobility and Moran's I [45] as the quantitative measurement. Given a set of features and an associated attribute, Moran's I and z-scores can be calculated to evaluate whether and to what degree the pattern expressed is clustered.

To analyze the spatial autocorrelation, a spatial fishnet was created first and the geo-tagged posts were then mapped into the corresponding fishnet cells, according to the post locations. Let $T$ be the collection $T = [t_1, \ldots, t_m]$ of continuous time segments $t_i$, and $C$ be the collection $C = [c_1, \ldots, c_n]$ of the fishnet cell $c_i = (r_i, num_i(t_k))$, where $r_i$ is the spatial region of $c_i$, and $num_i(t_k)$ is the number of the posts $p_j$, where $p_j.l_j \in c_i.r_i$ AND $p_j.t_j \in t_k$ AND $p_j.tps_j$ is of specific topics.

The Global Moran's I based on the post number can be further calculated as below [46]:

$$I_{t_k} = \frac{\text{n}}{S_0} \frac{\sum_{i=1}^{n} \sum_{j=1}^{n} w_{ij} z_i z_j}{\sum_{i=1}^{n} z_i^2} \tag{5}$$

where $z_i$ is the deviation of $c_i. num_i(t_k)$ from its average value $(c_i. num_i(t_k) - \frac{\sum_{j=1}^{n} c_j. num_j(t_k)}{n})$, $w_{ij}$ is the spatial weight between $c_i.r_i$ and $c_j.r_j$, and $S_0$ is the summation of all the spatial weights.

The value of Moran's I ranges from [–1,1]. The more similar values cluster, the closer $I$ is to +1; the more dissimilar values cluster (similar values disperse), the closer $I$ is to −1; Moran's I being +1 indicates perfect clustering of similar values while Moran's I being −1 indicates perfect dispersion. To construct the event feature, we develop a global event index (*GEI*) as below:

$$GEI_{t_k} = -c \cdot ln(1 - | I_{t_k} |) \tag{6}$$

where c is a scale coefficient for normalization. By using Equation (6), the *GEI* value will acceleratedly increase as $I_{t_k}$ approaches +1 (−1), indicating the transition from random to geographical clustering (dispersion).

By sequentially sampling *GEI* in the $T = [t_1, \ldots, t_m]$, a discrete time series will be generated:

$$TS = [GEI_{t_1}, GEI_{t_2}, \ldots, GEI_{t_m}] \tag{7}$$

where each item is a transformed feature for event representation in the created one-dimensional feature space. Different temporal granularity for analysis can be obtained, by adjusting the span of time segment $t_i$.

### 3.4. Detection of Event Feature Irregularity

Here, we aimed to detect irregularity of event features and pinpoint the spatiotemporal identification of the social event. Thus, for irregularity detection, we adopted the generalized extreme studentized deviate (ESD) test [47], which is a statistical test to detect one or more outliers in a univariate data set. The time series $TS$ (see Equation (7)) will be the input data set for the test and the outliers of the $TS$ will be found accordingly. The ESD test is explained in detail in Algorithm 1.

$RFunc(X)$ finds the index $k$ that maximizes the value of $R_k = \frac{|x_k - \bar{x}|}{s}$, where $x_k$, $\bar{x}$, $s$ denote the $k^{th}$ item, mean value and sample standard deviation of dataset $X$, respectively. The $Lambda(i)$ are calculated as follows:

$$Lambda(i) = \frac{(n - i)t_{p, n-i-1}}{\sqrt{(n - i - 1 + t_{p,n-i-1}^2)(n - i + 1)}} \tag{8}$$

where $n$ is the item count of dataset X, $t_{p,v}$ is the 100p percentage point from the $t$ distribution with $v$ degrees of freedom and $p = 1 - \frac{\alpha}{2(n-i+1)}$, with $\alpha$ being the significance level for the ESD test.

Apart from temporally detecting the happening of an event with the above ESD test, another important issue is where the event happened. To further identify the event location, the *GEI* is localized to construct a local event index (*LEI*), as below:

$$LEI_{t_k}(c_i) = -ln(1 - c|I_{t_k}(c_i)|) \tag{9}$$

where $LEI_{t_k}(c_i)$ is the calculated *LEI* for the fishnet cell $c_i$ in the time segment $t_k$, $I_{t_k}(c_i)$ is $c_i$'s Local Moran's I [46] calculated from $c_i. num_i(t_k)$ is the attribute value, and c is a scale coefficient

for normalization. If the fishnet cell has a high LEI this indicates particular geographical clustering or dispersion patterns of human mobility, which may be caused by special social events in that region.

By adopting the workflow described above, a potential social event can be detected by the identification of the semantic topics and the depiction of underlying spatiotemporal patterns.

---

**Algorithm 1.** Generalized Extreme Studentized Deviate Test

---

**Input**: (1) Observation dataset $X$ (2) Upper Bound Count $r$
**Output**: A list of detected outliers $L$

---

**Begin**
    **for** $i = 1$ **to** $r$ **do**
        $(R_k, k) \leftarrow RFunc(X)$
        $\lambda_i \leftarrow Lambda(i)$
        **if** $R_k > \lambda_i$ **do**
            $L.\,\mathrm{add}(X.\,\mathrm{item}(k))$   // The $k^{th}$ item in $X$ is added to $L$
        **end**
        $X \leftarrow X.\mathrm{item\_remove}(k)$   // Remove the $k^{th}$ item from $X$
    **end**
    Return $L$ // Return the list of detected outliers
**End**

---

## 4. Experiments

In this section, we implemented our proposed workflow and report our experiments with a real-world dataset.

### 4.1. Experimental Settings

The check-in dataset used for the experiment was retrieved from Instagram API, which covered 127,630 geo-tagged photos in Hong Kong from 6 December 2014 to 4 January 2015, generated by 26,420 users. A social media post included the post ID, user ID, the attached hashtags, the time-stamp of the online post and the location (longitude and latitude) indicating post place (Table 1). All the user IDs were anonymized for privacy protection.

**Table 1.** Description of social media post field.

| Fields | Description |
| --- | --- |
| pid | A string uniquely indicating a post |
| uid | An encrypted string uniquely indicating a user |
| hashtags | The attached hashtags indicating the potential post topics |
| stime | The time that the user publishes the online post |
| location | A pair of longitude and latitude indicating the post location |

Some of the user accounts were for commercial advertising and might post redundant photos in specific venues. Consequently, an anomaly detection method was used to remove these outliers. The number of photos posted by each user was first investigated. The calculations showed that the average count of posted photos per user $\mu$ was 4.8 and the standard deviation $\sigma$ was 8.8. The three-sigma rule [48] was introduced, which states that, for both normally distributed and non-normally distributed variables, most cases should fall within three-sigma intervals. Therefore, we used three-sigma intervals to set the threshold for filtering. The users whose photo numbers were outside the three-sigma intervals (i.e., $\mu + 3\sigma$, 32) were recognized as outliers and their posted photos were removed from the photo dataset. After data cleaning, 104,366 photos generated by 25,996 users

remained. For the parameter setting, the span of time segment $t_i$ was set as 24 h and the size of fishnet cell $c_i$ was set as $100 \times 100$ m.

### 4.2. Experimental Results

The experimental results were reported and the performance of our workflow was evaluated accordingly. Evaluating event detection techniques is a very challenging and complex problem, and there is still no commonly accepted standard [3]. Theoretically, the results of event detection can be evaluated by quantitative and qualitative analysis. In practice, quantitative evaluation requires a thorough ground truth data set, which is always unavailable and undermines the effectiveness and even feasibility of this evaluation approach. Therefore, a qualitative evaluation was adopted in our experiment.

#### 4.2.1. Event Detection by Feature Irregularity

After completing the semantic community discovery, several hashtag groups were detected, each of which shared a common topic. To demonstrate the usefulness of our workflow, we chose two hashtag groups for case studies.

We found one hashtag community whose semantic topic was mainly about the umbrella movement (see Figure 4). The umbrella movement was a political movement that emerged during the Hong Kong democracy protests of 2014 [49]. This movement lasted for 79 days, and many downtown streets were occupied during the event, which ended in a police clearance operation. By studying the spatiotemporal patterns of the corresponding geo-tagged check-ins, we are able to look into the event from a new perspective.

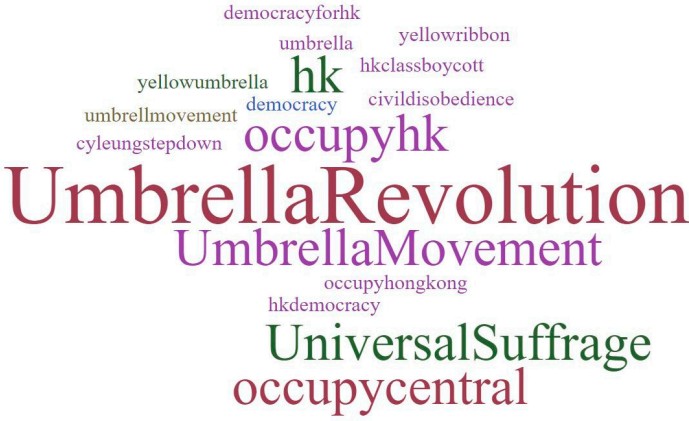

**Figure 4.** Word cloud for representative hashtags of the 'Umbrella Movement' community.

The *GEI* was calculated with the geo-tagged check-ins that were identified as the 'Umbrella Movement' topic. Table 2 lists the distribution of time series *TS* of *GEI*. Each *GEI* in time series *TS* represented the corresponding event feature for that time segment (i.e., that day). By implementing the ESD test upon the *TS*, the irregular event feature was detected. The visualized results are shown in Figure 5. It can be seen that at the beginning, the *GEI* value increased in an oscillatory manner and achieved its maximum of 37.43 on 11 December 2014, which was detected as the irregularity of the event features by the ESD method, and then decreased to its normal level after that.

**Table 2.** Time series of *GEI* with geo-tagged check-ins identified as the 'Umbrella Movement'.

| Date | Dec 6 | Dec 7 | Dec 8 | Dec 9 | Dec 10 | Dec 11 | Dec 12 | ... | Jan 4 |
|------|-------|-------|-------|-------|--------|--------|--------|-----|-------|
| *GEI* | 3.13 | 3.72 | 1.48 | 13.18 | 8.13 | 37.43 | 6.48 | ... | 0.61 |

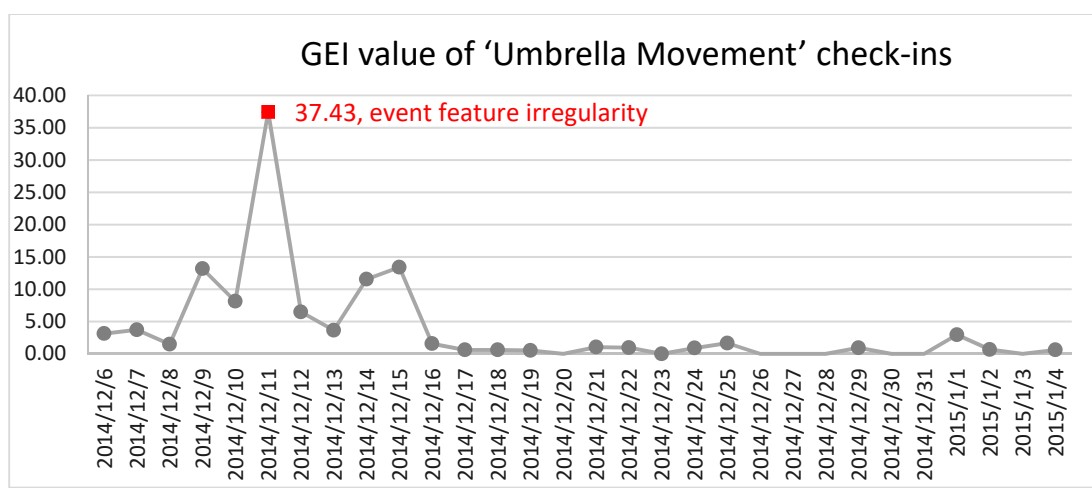

**Figure 5.** Detected feature irregularity as the 'Umbrella Movement' event on 11 December 2014.

Moreover, to identify the potential event location, the *LEI* (see Equation (9)) was also calculated for 11 December 2014. The spatial distribution of *LEI* is shown in Figure 6. We found that the regions with a high *LEI* value were mainly around Admiralty, where the key governmental departments (i.e., the Hong Kong Central Government Offices, Legislative Council Complex and the High Court of Special Administrative Region) are located, and the place with the highest *LEI* is at [lat: 22.28, lon: 114.17], where the *LEI* are 125.46, which according to Equation (9), indicates significant human aggregation in that region and the regions nearby. To determine the underlying reason causing this human movement pattern, we searched the background news with reference to the semantic identification 'Umbrella Movement' and detected the spatiotemporal pattern, 'Admiralty, 11 December 2014' (see Equation (1)). By researching the local news, we identified the relevant event, the Admiralty Site Clearance [50]. On Thursday 11 December 2014, the Hong Kong police executed the first site-clearance operation to end the main sit-in of the 'Umbrella Movement' in Admiralty with the arrest of 247 people. The police released an announcement about the site-clearance before the operation. Therefore, many citizens went to the Admiralty site to witness this social event on that day, causing significant human aggregation in Admiralty, which consequently was indicated by the *LEI* spike in the corresponding regions (Figure 6).

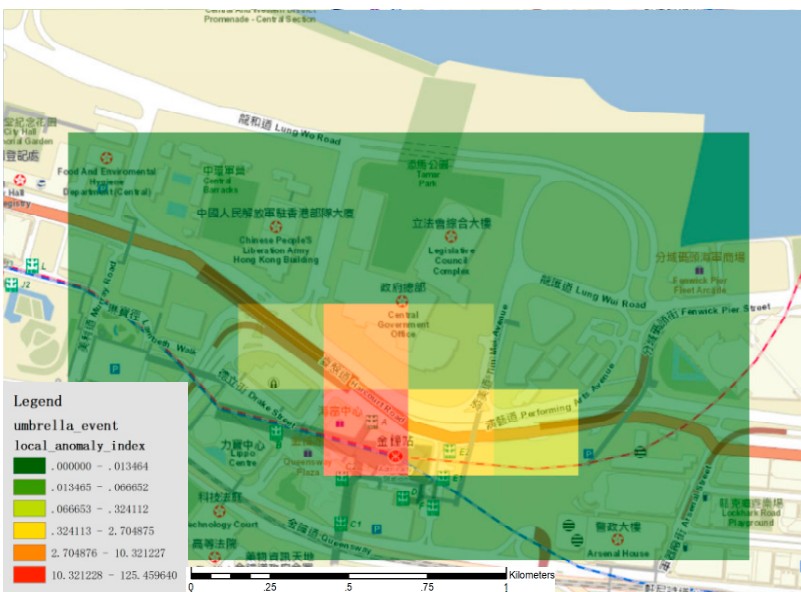

**Figure 6.** The Admiralty region, which shows high *LEI* values for 'Umbrella Movement' check-ins, indicating significant human aggregation in that region and the regions nearby.

To further demonstrate the effectiveness of our workflow, another case study is described below. We found another hashtag community whose semantic topic is mainly about fitness. The word cloud for the hashtags of that group is shown in Figure 7.

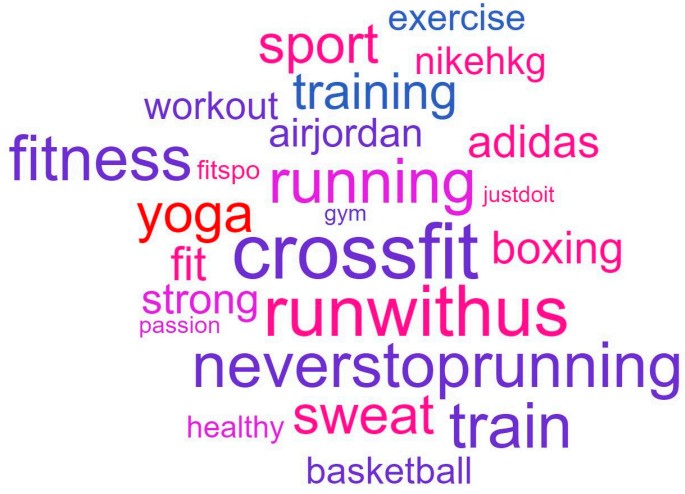

**Figure 7.** The word cloud for representative hashtags of a 'fitness' community.

We calculated the *GEI* with the geo-tagged check-ins identified as 'fitness' (see Table 3) and detected the event feature irregularity with the ESD test (see Figure 8). The event irregularity was detected on 7 December 2014, when the GEI value achieved its maximum of 32.47.

**Table 3.** Time series of *GEI* with geo-tagged check-ins identified as 'fitness'.

| Date | Dec 6 | Dec 7 | Dec 8 | Dec 9 | Dec 10 | Dec 11 | Dec 12 | ... | Jan 4 |
|------|-------|-------|-------|-------|--------|--------|--------|-----|-------|
| *GEI* | 1.20 | 32.47 | 1.77 | 0.86 | 0.34 | 0.35 | 0.76 | ... | 0.01 |

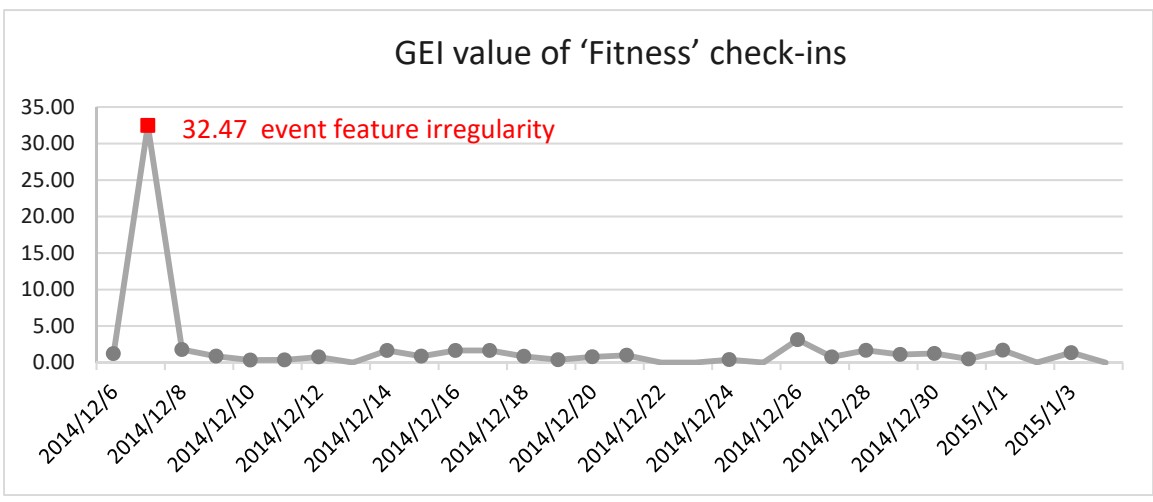

**Figure 8.** The detected feature irregularity of the 'fitness' event on 11 December 2014.

The location information was needed to pinpoint the social event; therefore, the *LEI* wascalculated. Figure 9 shows the spatial distribution of *LEI* calculated from the 'fitness' geo-tagged check-ins on 7 December 2014. The results showed that some areas in Chek Lap Kok had unusually high *LEI* compared with other regions. The area with the highest *LEI* is at [lat: 22.32 lon: 113.94], where the *LEI* was 189.23. This area is actually the AsiaWorld-Expo, which has the biggest purpose-built indoor-seated entertainment arena in Hong Kong. We tried to find the underlying event related to

this irregularity of *GEI* and *LEI*, by combing the semantic identification 'fitness' and spatiotemporal pattern 'AsiaWorld-Expo, 7 December 2014' (see Equation (1)). After searching the relevant background information, we identified the Color Run Hong Kong event [51]. This event was the first Color Run event in Hong Kong and was held on 7 December 2014 at the AsiaWorld-Expo. This was one of the most popular single day events for the Color Run in the Asian region, and there were approximately 16,000 fitness fans and runners participating in this event, bringing about immense human aggregation to the AsiaWorld-Expo, which was reflected by the detected irregularity in the calculated *GEI* and *LEI* (Figures 8 and 9).

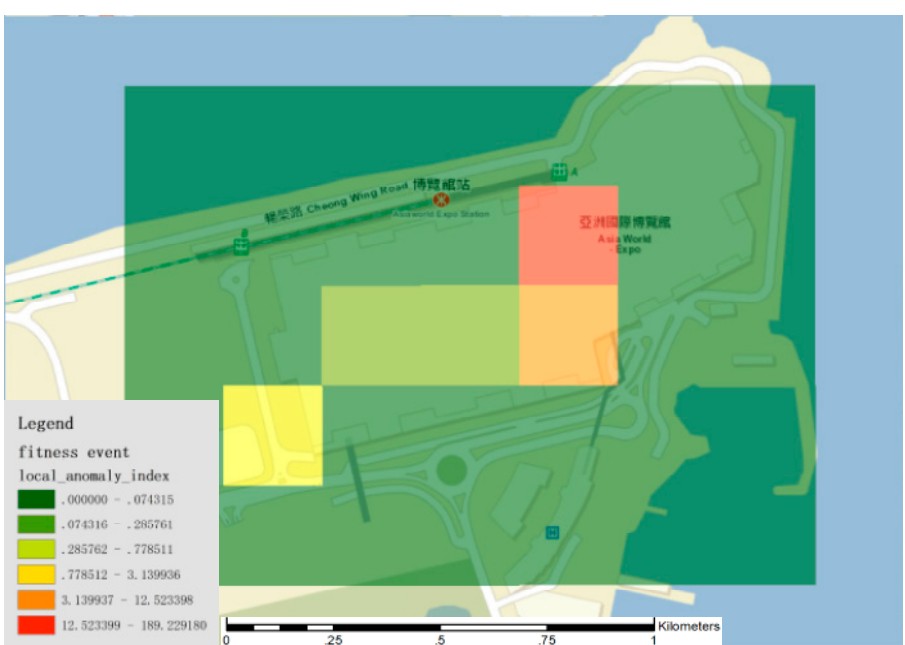

**Figure 9.** The Chek Lap Kok region, which shows high *LEI* values for 'fitness' check-ins, indicating significant human aggregation.

## 4.2.2. Urban Structure Understanding

Our workflow can also be used for studying the urban structure. Studying the urban structure on a large scale has traditionally been a challenge, requiring considerable labor and often resulting in a partial depiction of reality. We speculated that the geo-tagged check-ins generated by residents reveal the patterns of human mobility and aggregation in an urban context, which can be used for studying the structure and composition of a city. An application scenario is the study of human mobility during a festival. How people move and interact within the urban region during a festival can reveal the distinctly functional areas of the city and residents' perception about them. Consequently, in our experiment, the *GEI* and *LEI* for a festival event were calculated. One semantically detected hashtag community is about 'Christmas' (Figure 10).

To investigate the geographical patterns of human mobility and aggregation during Christmas, we calculated the *LEI* with the 'Christmas' geo-tagged check-ins on 25 December 2014. In Figure 11, we can see, across the whole urban area of Hong Kong, the regions with high *LEI* were mainly located within three regions, Tsim Sha Tsui, Lan Kwai Fong and Causeway Bay. This finding is consistent with the urban structure of Hong Kong. All three regions are major recreational areas that residents visit for shopping and entertainment during the holidays. Specifically, the regions with the highest *LEI* were around Lan Kwai Fong, which is the most well-known area in Hong Kong for drinking, clubbing and dining. This can be explained by the fact that, in Lan Kwai Fong, various street performance and celebration activities are held every Christmas, which would attract many tourists and local residents to visit, thus, causing extreme human aggregation in that region.

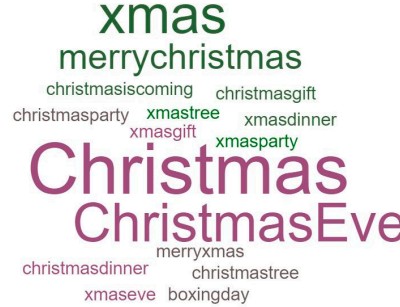

**Figure 10.** The word cloud for representative hashtags of the 'Christmas' community.

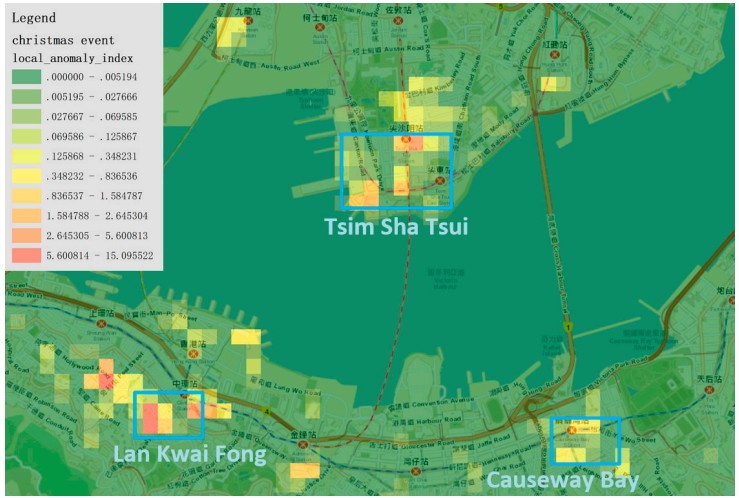

**Figure 11.** The *LEI* values across the Hong Kong urban area, calculated from the 'Christmas' check-ins. The high-value regions are mainly in Tsim Sha Tsui, Lan Kwai Fong and Causeway Bay, which are all major recreational regions in Hong Kong.

## 5. Discussion

As illustrated in Equation (1), to detect potential social events, our analytical workflow mainly considers two issues, semantic identification and geographical patterns, revealed by VGI data. In this section, we discuss the results of our methodology with the aim of answering the following questions:

- **Semantic Representativeness**: How well could the semantic similarities between hashtags be represented by the connectivity between vertices in a hashtag network? In other words, from a semantic perspective, were the detected semantic communities internally consistent and externally different?
- **Geographical Event Depiction**: How did we depict the social event from a geographical perspective? How were such geographical patterns, in turn, used for constructing event features and finally, for detecting events?

The effectiveness and limitations of our methodology are discussed using both quantitative and qualitive analysis.

### 5.1. Semantic Representativeness

We implemented Google *Word2vec* to investigate the semantic similarities between hashtag communities. *Word2vec* is a group of neural network models that take a large corpus of text as training data and produces a multidimensional vector space, where each unique word in the corpus is represented by a corresponding vector. Words that share common contexts are located in close proximity to one another in the vector space and have high cosine similarity [52]. The corpus of the

texts attached to geo-tagged photos was input as training data to create a vector space. Let $C_m$ and $C_n$ be two detected hashtag communities, the community semantic similarity (abbreviated as CSS) between $C_m$ and $C_n$ is calculated as below:

$$CSS(C_m, C_n) = \frac{\sum_i \sum_j sim\left(tag_i^m, tag_j^n\right)}{\text{Tg\_Cnt}(C_m) * \text{Tg\_Cnt}(C_n)} \tag{10}$$

where $tag_i^m$ is the $i^{th}$ tag in $C_m$, $sim(tag_i^m, tag_j^n)$ is the cosine similarity between $tag_i^m$ and $tag_j^n$ in the *Word2vec* vector space, and $\text{Tg\_Cnt}(C_m)$ is the total count of the tags of $C_m$. The higher the value of *CSS*, the more similar the hashtag communities are from a semantic perspective.

The divisions of hashtag communities were visualized using a comparative word cloud of the corresponding representative hashtags (shown in Figure 12).

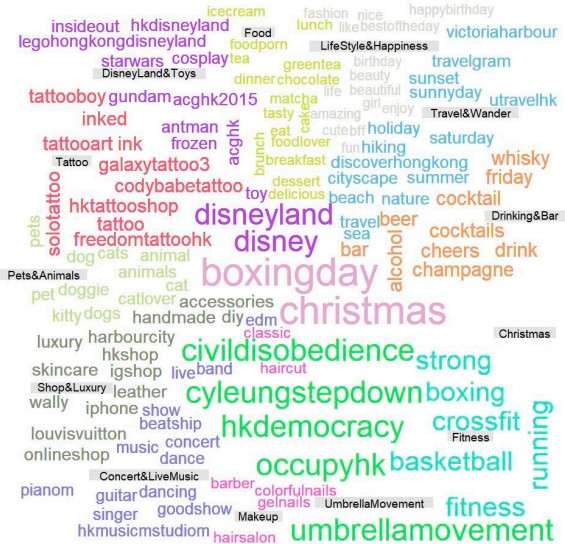

**Figure 12.** A comparative word cloud of 13 hashtag communities.

Table 4 showed the semantic similarities between hashtag communities. For hashtag communities $C_n$ we introduced intra community semantic similarity (*intra-CSS*) calculated by $CSS(C_n, C_n)$ (see Equation (10)) to measure the internal semantic consistency within the hashtag community $C_n$ and inter community semantic similarity (*inter-CSS*) calculated by $CSS(C_n, C_m)$ ($n \neq m$) to measure the semantic differences between communities $C_n$ and $C_m$. Semantically, high *intra-CSS* indicates that the detected community is internally consistent; significant differences between *intra-CSS* and *inter-CSSs* indicate that the detected communities are externally differentiable.

We listed the maximum, minimum and average values of *inter-CSS* together with *intra-CSS* for comparison. The results show that the detected communities had very high *intra-CSS* (approximately 0.90). The highest *intra-CSS* were achieved when it came to community 'Tattoo', which also achieved a low value in average *inter-CSS* (0.30). By studying the text content and users in detail, we found that this hashtag community was mainly popular among a population with a strong specific interest. When they posted photos of relevant topics on social media, the attached hashtags were always very activity-oriented and even specialized, which means the hashtags in these communities were mainly related to very specific activities, i.e., tattoos, and the chances that such hashtags co-occurred with the hashtags from other communities was relatively low. For example, hashtag 'solotatoo' from the 'Tattoo' community, might occur frequently with 'ink', another popular hashtag from the 'Tattoo' community, but seldom co-occurred with the hashtags from other semantic communities such as 'Food' or 'Pets&Animals'. Consequently, such peculiar co-occurrence patterns contributed to stronger internal, rather than external community connectivity, leading to high *intra-CSS* and low *inter-CSS*. A similar explanation can also be applied for another relevant phenomenon, i.e.,

the 'LifeStyle&Happiness' community had a relatively high *inter-CSS* value, indicating a strong external connectivity. A potential reason was that the hashtags in 'LifeStyle&Happiness' were mainly emotion-related (e.g., 'amazing', 'happy', 'enjoy'). These emotion-related hashtags tended to co-occur with other activity-oriented hashtags to express a user's emotional reactions and evaluations of the activities, leading to communities with more external connectivity and a higher *inter-CSS* value. Among all the communities, the *intra-CSSs* were mostly achieved with a very high value (over 0.85) and the differences between *intra-CSSs* and average *inter-CSSs* were mostly over 0.5 (the exception was 'LifeStyle&Happiness', whose difference was 0.44). The results verified that the division of semantic communities was internally consistent and externally different.

**Table 4.** A comparison of the semantic similarities between communities.

| Hashtag Communities | Intra-CSS | Inter-CSS | | |
| --- | --- | --- | --- | --- |
| | | Maximum Inter-CSS | Minimum Inter-CSS | Average Inter-CSS |
| Drinking&Bar | 0.89 | 0.56 | 0.32 | 0.38 |
| Travel&Wander | 0.93 | 0.61 | 0.26 | 0.38 |
| LifeStyle&Happiness | 0.92 | 0.69 | 0.37 | 0.48 |
| Food | 0.91 | 0.63 | 0.16 | 0.41 |
| DisneyLand&Toys | 0.93 | 0.54 | 0.22 | 0.43 |
| Tattoo | 0.95 | 0.38 | 0.23 | 0.30 |
| Pets&Animals | 0.94 | 0.57 | 0.20 | 0.43 |
| Shop&Luxury | 0.93 | 0.58 | 0.11 | 0.38 |
| Concert&LiveMusic | 0.93 | 0.59 | 0.26 | 0.40 |
| Makeup | 0.92 | 0.57 | 0.26 | 0.36 |
| Fitness | 0.86 | 0.56 | 0.32 | 0.37 |
| Christmas | 0.90 | 0.37 | 0.26 | 0.28 |
| UmbrellaMovement | 0.84 | 0.39 | 0.12 | 0.22 |

## 5.2. Geographical Event Depiction

To summarize, to detect social events from geo-tagged social media data, three aspects were considered by our data model. First, our consideration was that the semantic meaning should be investigated to identify the topics for social events. Therefore, we adopted a data-driven topic modeling method to detect hashtag communities and identify potential topics for the posts, in the rich-hashtag environment. Second, besides semantic identification, spatiotemporal identification was also considered by our model to pinpoint the events. This was derived from the perception that across an area, activities in the same category may take place at different places and times, which means that only by combining semantic and spatiotemporal identification can we specify a particular event. Consequently, in our model, a comprehensive workflow was implemented that included both global and local indexes to indicate the temporal and spatial information of the events, respectively. Third, to differentiate social events, we developed a novel geography-based event representation method. Previous works have mainly detected events by investigating the semantic feature of the streaming social media data, while we considered that social events may also lead to irregular geographical patterns in human mobility, and such geographical irregularity, in turn, can be used to detect social events. Specifically, we constructed a one-dimensional event feature based on the quantitative measurement of crowd aggregation (spatial autocorrelation) and detected the event by finding the feature irregularity. By experimenting with sample data, several events (e.g., a site-clearance operation and Color Run) were detected. We found that, as a matter of popularity, those events attracted a large number of citizens to the site, causing unusual human aggregation in corresponding areas, and subsequently, event feature irregularity. Such irregularities were captured and finally, the events were detected using our method.

## 5.3. Limitations and Future Work

Detecting social events with VGI data in this research has certain limitations. One limitation is that the categories of geographical patterns that we considered to construct event features are still

relatively simple and lacking in variety. In this study, we picked the clustering patterns (e.g., spatial autocorrelation) and used a logarithmic formula for event transformation. This is an initial attempt, however, it proves the feasibility of detecting events using geography-based features. We hope to extend the current work into a comprehensive analytic framework where different kinds of geographical pattern indicators and feature transformation operations are tested and incorporated so that social events can be more effectively differentiated. Another limitation is that the evaluation methods are still limited in their effectiveness. In our experiments, we did find that there were small parts of the irregularities that could not be meaningfully interpreted due to a lack of relevant background information. However, these only accounted for a very minor portion of the total detected events. Most of the detected irregularities, when combined with the semantic and spatiotemporal identification (Equation (1)), could be clearly interpreted and assigned to the corresponding social events. Due to the unavailability of a thorough ground truth data set, the current evaluation of the detection results was conducted using qualitative methods, and quantitative evaluations are absent. An authoritative ground truth dataset can be very helpful in achieving plausible quantitative evaluations. Finally, using social media data to detect events inevitably suffers from data bias issues because social media users are a skewed sample of the entire population, mainly consisting of specific groups of people and the younger generation. How to quantify and alleviate the influence of data bias and improve the detection results could be a potential future research direction.

## 6. Conclusions

Event detection with social media is not a novel topic and several relevant studies have been done. Some previous methods have been very effective. However, the novelty of our work lies in providing new insights into the following issues: what kinds of features can be used for detecting events, or, from what aspects can a social event be differentiated using VGI data. Previous methods are mostly semantic-based, which is to say they detect the "bursts" of certain semantic signals to identify events. By contrast, our consideration is that a social event will also affect how the objects spatially distribute and mutually interact, thus causing irregular geographical patterns. Furthermore, by introducing depictive features with VGI, such geographical irregularities can be quantified and used to distinguish social events. Based on the above consideration, we designed a comprehensive workflow where the event feature was represented by investigating the geographical pattern (e.g., spatial autocorrelation) of VGI and the social events were thereby detected.

The proposed workflow consists of three modules: (1) semantic community discovery, (2) geography-based event representation, and (3) detection of event feature irregularity. First, we implemented a data-driven geographic topic modeling method to detect hashtag communities and identify social media topics. Second, by introducing quantitative spatial autocorrelation indicators, a global index was constructed for representing the potential events in the created feature space. A time series of the univariate event feature was then generated with variable temporal granularities. Third, an outlier test was adopted to detect event feature irregularity, and the global index was localized for finding event location.

The main innovations in our work lies in constructing event features based on the geographical patterns revealed by VGI and differentiating events by detecting feature irregularities, which are mainly related to the second and third modules (i.e., geography-based event representation, and detection of event feature irregularity) of our proposed workflow, where both global and local indicators (features) are constructed by investigating the geographical patterns of VGI data and feature irregularities are detected. By experimenting with sample datasets, this study demonstrated the feasibility of detecting social events with geography-based features, and it could potentially complement current semantic-based event detection methods, which is the scientific weight of our study. Our current work is a preliminary workflow; nevertheless, it provides a framework for future work. Detecting social events by mining geographical irregularities is a potentially promising direction, as relatively few

relevant works have done this previously, compared with the massive number of semantic-based event detection methods.

**Author Contributions:** Z.L. conceived the methodology and conducted the experiments. Z.L. and X.Z. designed the evaluation. W.S. identified the problem and solution. Z.L. in collaboration with A.Z. wrote the manuscript.

**Funding:** This study is funded by The Hong Kong Polytechnic University [1-ZVF2, 1-ZEAB]

**Acknowledgments:** We would like to thank the anonymous reviewers for their insightful comments and substantial help on improving this article. We also thank Jittin Chaitamart for providing the valuable sample data.

**Conflicts of Interest:** The authors declare no conflicts of interest.

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
