# Peer review of "Towards Detecting Social Events by Mining Geographical Patterns with VGI Data"

_ijgi, doi:10.3390/ijgi7120481_

Round 1
Reviewer 1 Report
This paper examined geographic topic discovery and event detection within social media data using pattern-mining techniques. The authors proposed a novel a workflow to detect emerging social events by investigating the geographical patterns of VGI data. They evaluated their approach within several contextual setting and geographic areas of Hong Kong.
The introduction and literature review are very good in providing the background for this topic and in providing the rationale for this study. The discussion and conclusion section is well-written and identifies the strengths and weaknesses of this research project stating that that the success of this approach depends on an authoritative ground-truthed dataset so that a rigorous assessment can be performed (the authors indicate that they lacked such a dataset). In addition, they point out that using social media data to detect events inevitably suffers from data bias issues (“social media users are a skewed sample from the whole population, mainly consisting of specific groups of people and young generation”). These are not new issues or discoveries, but are useful to state nonetheless.
The Analytic Workflow presented in Figure (1) should be useful to other researchers in helping them to formulate additional research streams related social media / geographic pattern-mining techniques in general, and this topic specifically.
Overall the paper is well-written, though, a general, re-read / editing session of the document would be useful as there are some grammatical errors, (e.g., sentences ending with a preposition), and the use of the first person narrative - though lately, these have become more acceptable in some publications. In addition, most of the images are a bit blurry (it would be great to include higher-resolution images). However, these are minor issues and I believe this is a very useful contribution to the GIS field in general, and to the VGI domain in particular.
Author Response
1.Overall the paper is well-written, though, a general, re-read / editing session of the document would be useful as there are some grammatical errors, (e.g., sentences ending with a preposition), and the use of the first person narrative - though lately, these have become more acceptable in some publications. In addition, most of the images are a bit blurry (it would be great to include higher-resolution images).
Response: Thanks for your regards and suggestions. We have made a thorough proof-reading and revised some unproper grammatical errors.
Besides, we have also added some new contents to emphasis the originalities, contributions and potential improvements of our work. Please see the highlighted parts of the new manuscript for details.

Reviewer 2 Report
This article outlines a method to extract events from geo-located social media. The method combines the analysis of some semantic content (hashtags) and locational content to identify high-density areas (spatially, thematically, and temporally). The article is fairly well-written and clear. While the work definitely has some merit, I think it needs major revisions before potential publication:
- The authors claim that event detection is still hard, but this seems an exaggeration, particularly for the kind of events that the authors refer to in their evaluation. More discussion and evidence is needed to convince the reader that this sort of detection has not been tried before (see additional references).
- Rather than creating a new method, the article seems to combine existing, well-known methods, including topic models, spatial clustering techniques, word2vec, etc. Moreover, the way the methods are combined looks fairly simple and unsophisticated. This might be very good for implementation purposes in some real systems, but it lacks scientific weight. The article should clarify very openly what is original in this approach. Notably, there is a massive literature on Twitter event and topic detection, and it seems unlikely that this article contains something original in this respect.
- The evaluation is conducted on a small dataset on a few cherrypicked case studies, and hence not very convincing, as it focusses on large events with clear spatio-temporal scopes that look therefore quite easy to identify (a large protest, a marathon, etc.). Furthermore, it is unclear how much noise (i.e. uninterpretable events) the method identifies in the data. The evaluation (100K Instagram photos in HK) should be mentioned openly in the abstract and introduction.
- More references to existing work on semantic similarity are needed. More broadly, relevant references that should be discussed to justify the work's originality:
Becker, H., Naaman, M., & Gravano, L. (2011). Beyond Trending Topics: Real-World Event Identification on Twitter. Icwsm, 11(2011), 438-441.
Panteras, G., Wise, S., Lu, X., Croitoru, A., Crooks, A., & Stefanidis, A. (2015). Triangulating social multimedia content for event localization using Flickr and Twitter. Transactions in GIS, 19(5), 694-715.
Abdelhaq, H., Sengstock, C., & Gertz, M. (2013). Eventweet: Online localized event detection from twitter. Proceedings of the VLDB Endowment, 6(12), 1326-1329.
Ritter, Alan, Oren Etzioni, and Sam Clark. "Open domain event extraction from twitter." Proceedings of the 18th ACM SIGKDD international conference on Knowledge discovery and data mining. ACM, 2012.
Author Response
Response to Reviewer 2:
Thank you so much for your insightful comments and suggestions for improvement. We have revised our paper accordingly and responded each of your comments as below:
1.- The authors claim that event detection is still hard, but this seems an exaggeration, particularly for the kind of events that the authors refer to in their evaluation. More discussion and evidence is needed to convince the reader that this sort of detection has not been tried before (see additional references).
Response: Thanks for bringing up your concern. The major contribution of this article is to take advantage of new geographical features for event detection, which means using the change (“outliers”) of the geographical patterns revealed by VGI to differentiate potential events.
Yes, we agree that event detection with social media is not a novel topic and massive relevant works have been done. Some previous methods have shown good effectiveness. And we believe the kinds of events referred to in our study (i.e., political protest, ColorRun activity) can possibly be detected as well, by adopting some of previous methods. However, the points of our work lie in providing new insights into the following issues: what kind of features can be used for detecting events or, from what aspects, can a social event be differentiated with VGI data. Previous methods are mostly semantic-based (Weiler et al. 2016, Cordeiro and Gama 2016), which is to detect the “bursts” of certain semantic signals to identify events. While our consideration is that a social event will also affect how the objects spatially distribute and mutually interact, thus causing irregular geographical patterns, and by introducing depictive features with VGI, such geographical irregularities can be quantified and used to distinguish social events. Based on above consideration, we designed a comprehensive workflow where the event feature is represented by investigating the geographical pattern (e.g., spatial autocorrelation) of VGI and the social events are detected thereby.
After searching the relevant literature, we did find some previous methodologies that incorporated spatial-related components. However, these works still detected events mainly by finding burst of semantic signal, while the spatial-related components mainly served as a complementary role to identify the spatial information of the detected event, such as detecting event location (Sakaki et al. 2010, Paule et al. 2018) and assessing the influenced area (Abdelhaq et al. 2013, Panteras et al. 2015, Gao et al. 2018), rather than be used, per se, as major features to differentiate social events. Another previous study that might be potentially related to my work is Lee and Sumiya (2010), where the authors measured the regional regularities with tweet and user count, which is relatively simple and intuitive. While in our work, our consideration is that social events can not only lead to “bursts” of certain semantic signals and posts/users count, but also cause structural change of the geographical patterns like how objects spatially distribute and interact with each other. So, in our work, rather than investigating the numeric change of post/user count, we tried to construct features to indicate the structural change of objects’ geographical distribution pattern. Specifically, we picked up the clustering patterns (e.g., spatial autocorrelation) as a tentative attempt and designed the event features accordingly. The experiments were conducted and demonstrated the effectiveness of our design. To the best of our knowledge, detecting social events by mining irregularities of geographical patterns has not been adequately investigated by previous studies. And we think this work can shed some light on relevant fields.
In the newly submitted manuscript, we have cited your recommended works and emphasized the intuition of our work and our differences from previous works. (Abstract P1. L.14, Section 1 P.2 L.5 – L.13 ‘By introducing depictive …’, Section 2 P.2 L.41 L.46, P.3 L.7 L.15 L.21 L.26)
Abdelhaq, H., Sengstock, C. and Gertz, M. 2013. Eventweet: Online localized event detection from twitter. Proceedings of the VLDB Endowment, 6(12), 1326-1329.
Cordeiro, M. and Gama, J., 2016. Online social networks event detection: a survey. Solving Large Scale Learning Tasks. Challenges and Algorithms. Springer, 1-41.
Gao, Y., et al. 2018. Mapping spatiotemporal patterns of events using social media: a case study of influenza trends. International Journal of Geographical Information Science, 32(3), 425-449.
Lee, R. and Sumiya, K., Measuring geographical regularities of crowd behaviors for Twitter-based geo-social event detection. ed. Proceedings of the 2nd ACM SIGSPATIAL international workshop on location based social networks, 2010, 1-10.
Panteras, G., et al. 2015. Triangulating social multimedia content for event localization using Flickr and Twitter. Transactions in GIS, 19(5), 694-715.
Paule, J. D. G., Sun, Y. and Moshfeghi, Y. 2018. On fine-grained geolocalisation of tweets and real-time traffic incident detection. Information Processing & Management.
Sakaki, T., Okazaki, M. and Matsuo, Y., Earthquake shakes Twitter users: real-time event detection by social sensors. ed. Proceedings of the 19th international conference on World wide web, 2010, 851-860.
Weiler, A., Grossniklaus, M. and Scholl, M. H. 2016. Survey and experimental analysis of event detection techniques for twitter. The Computer Journal, 60(3), 329-346.
2.- Rather than creating a new method, the article seems to combine existing, well-known methods, including topic models, spatial clustering techniques, word2vec, etc. Moreover, the way the methods are combined looks fairly simple and unsophisticated. This might be very good for implementation purposes in some real systems, but it lacks scientific weight. The article should clarify very openly what is original in this approach.
Response: The main originalities of our work are to construct event feature based on the geographical patterns revealed by VGI and differentiate events by detecting feature irregularities. The originalities are mainly related to the second and third modules (i.e., Geography-based Event Representation, Detection of Event Feature Irregularity) of our proposed workflow, where both global and local indicators (features) are constructed by investigating the geographical patterns of VGI data and the feature irregularities are detected.
While designing the analytical workflow, our problem-solving principle is “Entities are not to be multiplied without necessity” (Occam's razor). We try to make each component clearly straightforward yet effective, rather than seemingly “sophisticated” or “fancy”. In this study, we did not seek to invent some novel topic modeling or outlier detection algorithms but focused on creating new geography-based features for event representation and detection. That’s why we adopted previously created methods for topic modeling, and outlier detection, because these served as a supporting role and were not the main focus (and intention) of our research. However, these previously created methods need to be effective in our workflow, so we conducted the experiments and evaluations (in section 4 and 5) and concluded that the effectiveness of the adopted methods can be (partially) demonstrated.
As mentioned above, the point of our study is to provide new insights into the following issues: what kinds of features can be used for event detection or, from what aspects, can a social event be differentiated with VGI data. Specifically, we tried to construct geography-based feature based on the consideration that a social event might cause irregular geographical patterns and such geographical irregularities could be used, in turn, to differentiate social events. By experiments with sample datasets, this study demonstrated the feasibility to detect social events with geography-based features and could be complementary to the current semantic-based event detection method, which is the scientific weight of our study. This can be a potentially promising direction as relatively fewer relevant works have been done before, compared with the massive volume of the semantic-based event detection methods.
On the other hand, we do agree that some parts of our proposed methodology can be further refined to be more “sophisticated” and “effective”. We think that one worthwhile question is what other geographical patterns and transformation formula can be used to create event feature so that the social events can be more differentiable in the new feature space. In this study, we picked the clustering patterns (e.g., spatial autocorrelation) and used logarithmic formula for event transformation. This is an initial attempt and lacking in varieties, but proves the feasibility of detecting events with the geography-based feature. And we hope to extend the current work into a comprehensive analytic framework where different kinds of geographical pattern indicators and feature transformation operations are tested and incorporated. We have mentioned this as one of our future directions (Section 6 P.17 L.29 ‘One limitation is…’).
In the newly submitted manuscript, we have also made several revisions to openly clarify the originalities and contribution of our methodology. Please see Section 1 P.2 L.10 ‘provide new insights …’, L.26 ‘Specifically, we presented …’, Section 2 P.3 L.26 ‘While in our work…’, Section 3 P.3 L.33 ‘Firstly, we introduced …’, Section 3.3 P.6 L.30 ‘we intend to construct…’, Section 5.2 P.16 L.33 ‘Thirdly, to differentiate…’ ,Section 6 P.17 L.25 ‘Our research examined…’.
Occam's razor [online]. Available from: https://en.wikipedia.org/wiki/Occam%27s_razor [Accessed 25 Nov 2018]
3.Notably, there is a massive literature on Twitter event and topic detection, and it seems unlikely that this article contains something original in this respect.
Response: Yes, we totally agree that there are plenty of previous study on event detection with social media data. However, most previous works used semantic-based methods, which are to detect events by finding burst of semantic signal, while detecting social events by mining irregularities of geographical patterns has not been adequately (if any) investigated. This can be a promising direction for future work, as relatively fewer relevant works have been done, compared with the massive volume of semantic-based methods. And we believe our research can serve as an exploration for the feasibility and shed some light on relevant fields.
The main contributions and originalities of this work has been more thoroughly elaborated in the above responses. Please refer to response 1 & 2 for details.
4.- The evaluation is conducted on a small dataset on a few cherrypicked case studies, and hence not very convincing, as it focusses on large events with clear spatio-temporal scopes that look therefore quite easy to identify (a large protest, a marathon, etc.).
Response: Our current work is a tentative attempt to detect social events with the geography-based feature. By looking into the detected events (e.g., political protest, marathon) in detail, we found that these events indeed brought change to the spatiotemporal patterns of the VGI and then cause irregular geography-based features, which were, in turn, used to differentiate the social events. As mentioned in response 1, we believe the events detected in our experiments could also possibly be detected by using previous well-established semantic-based methods. However, the originalities and points of our research are to detect social events by mining the geographical patterns of VGI and using geography-based features. And we think the current experiments, although admittedly limited, can demonstrate the feasibilities and effectiveness of this train of thought. Our current work is a preliminary workflow, nevertheless, it provides a framework for our future work. We hope more geographical pattern indicators and feature transformation operations can be incorporated into our analytical workflow, so that a more refined framework can be accomplished and some quantitative comparisons can be made to test the efficiency between the classical semantic-based methods and our methodology. (as mentioned in Section 4.2 P.9 L.9 ‘Evaluating event detection…’, Section 6 P.17 L.29 ‘One limitation is…’)
5. Furthermore, it is unclear how much noise (i.e. uninterpretable events) the method identifies in the data.
Response: Thanks for bringing up this very valuable issue. In our experiments, we did find that there were small parts of the irregularities can’t be meaningfully interpreted, due to lack of relevant background knowledge, however these only accounted for a very minor portion of the total detected events. Most of the detected irregularities, when combined with the semantic and spatiotemporal identification (Equation 1), can be clearly interpreted and assigned to the corresponding social events.
Admittedly, how to quantitatively evaluate the performance of event detection techniques is a very challenge and complex problem, inflicting most relevant study (our work is of no exception) and there is no commonly consented standard yet (Weiler et al. 2016). Theoretically speaking, to calculate the quantitative measurements (like precision, accuracy and recall), a thorough ground truth dataset is normally required, which is always unavailable in practice. We have admitted this lack of quantitative evaluation as a limitation of our current work and suggested introducing an authoritative ground truth dataset as a valuable direction of the future work. (See Section 6 P.17 L.35 ‘Due to the unavailability’)
We believe an authoritative ground truth dataset can be significantly beneficial to achieve plausible quantitative evaluations and can be a very valuable direction for our future work.
Weiler, A., Grossniklaus, M. and Scholl, M. H. 2016. Survey and experimental analysis of event detection techniques for twitter. The Computer Journal, 60(3), 329-346.
6.The evaluation (100K Instagram photos in HK) should be mentioned openly in the abstract and introduction.
Response: The dataset for evaluation has been mentioned in the new submission. (Abstract P.1 L.21, Section 1 P.2 L.22)
7.- More references to existing work on semantic similarity are needed.
Response: Thanks for your suggestions. More related works about topic modeling and natural language processing have been cited in the new submission. (Please see Section 2)
8.More broadly, relevant references that should be discussed to justify the work's originality: (Becker,…..)
Response: Thanks for providing the valuable materials. These recommended works have been cited in the new manuscript (See Section 2) and originalities of our work have been openly clarified (Section 1 P.2 L.10 ‘provide new insights …’, L.26 ‘Specifically, we presented …’, Section 2 P.3 L.26 ‘While in our work…’, Section 3 P.3 L.33 ‘Firstly, we introduced …’, Section 3.3 P.6 L.30 ‘we intend to construct…’, Section 6 P.17 L.25 ‘Our research examined…’).
Reviewer 3 Report
General Remarks:
This is an interesting and well-written paper which concerns detection of changes in geographic concentration of people by extraction and analysis of geo-tagged photography data obtained from the Instagram social network API. For this purpose, the authors have proposed a Global Event Index, which is based on spatial autocorrelation obtained by means of Moran's I. The proposed methodology is sound, and the presented results look convincing. I would recommend the article to be published following minor corrections.p { margin-bottom: 0.25cm; line-height: 120%; }
Detailed Recommendations:
Page 5 Lines 14-17: This sentence is almost a direct repetition of a sentence from section 2. It should be abbreviated or removed.
Page 7 Line 8: “approaches to +1” - “to” is unnecessary here.
Page 12 Line 10: Use “reveal” instead of “reveals”.
Page 15 Line 49: “While” is unnecessary in this context.
p { margin-bottom: 0.25cm; line-height: 120%; }
Author Response
1.Page 5 Lines 14-17: This sentence is almost a direct repetition of a sentence from section 2. It should be abbreviated or removed. (abbreviated)
Response: The sentence has been abbreviated to be concise.
2.Page 7 Line 8: “approaches to +1” - “to” is unnecessary here.
Response: The ‘to’ has been deleted.
3.Page 12 Line 10: Use “reveal” instead of “reveals”.
Response: The typo has been corrected
4.Page 15 Line 49: “While” is unnecessary in this context.
Response: The ‘while’ has been deleted.
Besides above revisions, we have also added some new contents to emphasis the originalities, contributions and potential improvements of our work. Please see the highlighted parts of the new manuscript for details.
Round 2
Reviewer 2 Report
The authors have kindly responded to my concerns in the letter, and I agree with most of their points. However, the editing in the actual article is very shallow. I think the authors should better integrate their responses into the article, particularly concerning the originality and limitations of the approach.
Author Response
Comments: The authors have kindly responded to my concerns in the letter, and I agree with most of their points. However, the editing in the actual article is very shallow. I think the authors should better integrate their responses into the article, particularly concerning the originality and limitations of the approach.
Response: Thanks so much for your suggestions. In the new submission, we have added contents to integrate our previous responses (particularly about the originalities and limitations of our work). Please see the highlighted sentences in the manuscript. (Section 1 P.2, Section 2 P.3, Section 5.3 P.17, Section 6 P.17-18)
Please be kindly noted that we have added a new section in 5.3 to more thoroughly discuss the limitations and future directions of our work.